# OpenReview forum: "CRMArena-Pro: Holistic Assessment of LLM Agents Across Diverse Business Scenarios and Interactions"
_TMLR — Accepted by TMLR_

### Review · Reviewer_XSGR · 2025-10-21

**Summary Of Contributions:**

The paper introduces *CRMArena-Pro*, a new benchmark for evaluating AI agents on business tasks. *CRMArena-Pro* expands CRMArena with multi-turn and business-to-consumer (B2C) tasks, and includes 4,280 queries for 19 different tasks covering customer service, sales, and CPQ (configure, price, quote) scenarios.

The paper evaluates strong LM agents based on OpenAI-o1 and Gemini-2.5-Pro, and shows that CRMArena-Pro is challenging, with these models reaching a performance of 49.5 and 58.3 in the single-turn setting, and 30.4 and 30.0 in the more challenging multi-turn setting.

Moreover, the paper analyzes the ability of agents to restrain from sharing confidential information, and shows this remains a significant challenge for future work, as models achieve near-zero performance without specialized prompting.

**Additional Comments:**

- Previous benchmarks such as τ2-Bench [1] also include intermediate metrics, such as natural language assertions that can be used to gain more insights on agents performance. Did you consider adding such metrics to *CRMArena-Pro*?

- CRMArena-Pro uses LM-simulators to simulate users -- what is the price for evaluating the benchmark with the LMs used in this work (Table 6)?

- It could be helpful to elaborate on the different tasks in the benchmark (Figure 3). For example, are Sales, Customer service, and CPQ tasks inherently different, and if so why?

[1] τ2-Bench: Evaluating Conversational Agents in a Dual-Control Environment, Barres et al., 2025.

**Audience:**

Yes

**Audience Explanation:**

Evaluation of LM agents remains a significant challenge. Researchers interested in evaluating agents on multi-turn, business tasks will be interested in *CRMArena-Pro*.

**Claims And Evidence:**

Yes

**Claims Explanation:**

The claims are mainly supported by the results presented in the paper. I do have some minor concerns which are detailed in the requested changes section.

**Requested Changes:**

> Additional models

While the paper experiments with OpenAI-o1 and Gemini-2.5-Pro and nine models overall, some LMs that are known to perform well in agentic settings are missing. For example, Claude models are known for their strong agentic performance, as are the newer o3, o4, and GPT-5 models from OpenAI, and Grok-code. For open-source models, Qwen3-coder and DeepSeek-R1 models could be an interesting addition. I believe the paper could benefit from experimenting with at least some of these models.

> Additional analysis

While the paper does provide interesting qualitative analysis, for example, regarding the use of different LM simulators, many details are not followed by additional quantitative human analysis. For example, the paper claims that it uses 21 latest variables to create realistic data. It could be interesting to also describe what makes the data realistic, and how this was verified. More importantly, it would be interesting to analyze the generated queries and agent trajectories. Does solving the tasks require diverse actions? What are the failure points of current agents? What kind of tasks are not reflected in *CRMArena-Pro*? How does Gemini-2.5-Pro perform so well on Workflow tasks (83%) yet is outperformed by o1 on multi-turn tasks? Providing additional human analysis that show the strenghts and limitations of the generated data and current agents can highly benefit future work.

> Additional information regarding data generation.

It would be helpful to elaborate on the pipeline used to generate the queries and answers, and adding examples in the paper (there are some examples for queries in the appendix).

> Non-realistic examples

The paper claims that around 1/3 of the data is not realistic (neutral or unrealistic). It was unclear to me why these queries were not removed from the final dataset.

> Confidentiality awareness

The paper claims that confidentiality awareness is a major challenge, yet it advocates for an LLM-as-a-judge approach for identifying when agents share confidential information. While the paper does discuss the fact this phenomenon can be mitigated with additional training, I believe it could benefit from empirically validating whether this is indeed the case, especially as confidentiality awareness is claimed as one of the main contributions of the paper (e.g., in the abstract and in Table 1).

Moreover, while the paper claims that a tradeoff between confidentially awareness and performance when using the "confidentiality-aware" prompt, the tradeoff is limited to certain LMs (e.g., performance for o1 is stable and Llama-4-Maverick seems to improve, Table 3).

---

> ### Author Response · Authors · 2025-12-12
>
> We thank reviewer XSGR for the recognition of CRMArena-Pro as a challenging benchmark that meaningfully expands upon prior work by introducing multi-turn interactions and B2C tasks across diverse business scenarios. We are also grateful for the reviewer's acknowledgement of our findings regarding the significant performance gaps in leading models like o1 and Gemini-2.5-Pro, particularly in multi-turn settings and confidentiality awareness. We have addressed the reviewer's thoughtful suggestions regarding additional model evaluations, qualitative analysis, and data validation in the response below.
>
>
>
> > Additional models
>
> We appreciate the suggestion to broaden our evaluation suite and we have integrated experiments for claude-sonnet-4.5 and qwen3-coder-480b-a35b into the revised manuscript. As shown in the updated Table 2, claude-sonnet-4.5 demonstrates exceptional capabilities, achieving approximately 89% success in single-turn Workflow Execution tasks, placing it in direct competition with gemini-2.5-pro. However, consistent with our broader findings, even this top-tier model experiences significant performance degradation in multi-turn settings (dropping to ~50%), further validating the rigorous nature of the CRMArena-Pro benchmark.
>
> Regarding open-source models, qwen3-coder-480b showed strong utility, outperforming the Llama 3.1 series in both B2B and B2C single-turn task completion. However, its multi-turn performance is significantly worse than other top-performing open-source models. As mentioned in our footnote 6, we excluded DeepSeek-R1 due to its sub-optimal performance on the original CRMArena tasks. We believe these additions provide a more holistic view of the current landscape, confirming that the challenges highlighted in our work remain unsolved across the strongest model families.
>
>
> > Additional analysis
>
>
> We have clarified that the 21 latent variables (e.g., churn_risk, product_usage_level) are not just random seeds but causal drivers that ensure logical consistency across the schema (e.g., a high churn_risk variable triggers a higher frequency of negative sentiment VoiceCallTranscript records and Case escalations). This causal consistency is exactly what makes the data "realistic" compared to random generation. This was verified not just by code checks, but through the Expert Studies (Section 3.4), where professionals validated that the data relationships mirrored their real-world CRM experiences.
>
> Regarding failure analysis, we have expanded Section 4.4 with a systematic manual analysis of 100 agent trajectories (50 single-turn, 50 multi-turn). We quantified specific failure modes, finding that 32% of single-turn errors stem from schema gaps (e.g., confusing PricebookEntry vs. OrderItem) and 45% of multi-turn failures arise from "premature finalization" (failing to ask for necessary missing info). This directly answers the question about failure points and confirms that solving tasks requires diverse actions—successful agents must dynamically switch between SOQL (structured), SOSL (unstructured), and clarification strategies.
>
> Regarding the performance dynamics, we clarify that Gemini-2.5-pro actually retains a lead in B2B multi-turn settings (35.1% vs. 32.5% for o1), though o1 is competitive in B2C. The dominance of Gemini-2.5-pro in Workflow Execution (83%+)  is attributed to the task nature: these tasks involve following explicit, rule-based policies (e.g., "If condition X, assign to Y") where Gemini's instruction-following is paramount.
>
> > Additional information regarding data generation.
>
> We appreciate the request for further details on the generation pipeline. Regarding the methodology, CRMArena-Pro strictly adheres to the data and query generation pipeline established in the original CRMArena (Huang et al., 2025). Specifically, queries and ground-truth answers are not post-hoc annotations but are programmatically derived from the latent variables and synthesized database state to ensure mathematical exactness.
>
> We have detailed the resulting environment and object schema in Appendix H, and provided comprehensive descriptions and example queries for every task in Appendix F (Figures 18–36). Due to space constraints and the high overlap with the prior work, we encourage readers to refer to Figure 1 of the original CRMArena paper for the visual schematic of the generative pipeline. We believe the combination of the detailed schema in Appendix H and the rigorous task definitions in Appendix F provides the necessary context for understanding the benchmark's construction.

---

> ### Author Response · Authors · 2025-12-12
>
> > Data realism clarification
>
> We wish to clarify a misunderstanding regarding the realism metric. The figure stating that roughly 67% of the data was rated "Realistic" or "Very Realistic"  refers to the proportion of expert ratings given to the Salesforce objects holistically, not a percentage of specific "bad" queries or records that can be isolated and removed. In our study, experts did not rate individual rows; instead, they browsed the records within specific objects (e.g., Accounts, Opportunities) and provided an overall rating for that object type based on their professional experience.
>
>
>
> Consequently, the remaining percentage does not represent a discrete set of "unrealistic" data points. The majority of these remaining ratings were "Neutral," defined in our study as "moderately realistic" (Appendix D). This reflects the subjective nature of the evaluation, where experts from different industries may have varying expectations for CRM data, rather than objective errors.
>
> > Confidentiality awareness LLM Judge reliability
> As discussed in Section 4.4, our evaluation is explicitly designed to be a constrained, binary classification rather than a subjective open-ended assessment. The judge is prompted with specific refusal phrases and instructed to output a simple "YES" or "NO" based solely on whether the agent explicitly refused the sensitive query. This reduces the potential for biases such as verbosity or self-enhancement often associated with LLM judges, making it a robust metric for this specific type of safety check.
>
> To empirically validate this reliability, we conducted a manual inspection of 50 randomly sampled judge outputs against the agent trajectories. Our manual evaluation identified only a single error, confirming that the judge aligns closely with human ground truth for this specific refusal-detection task.
>
> > Empirical validation of mitigation via training
>
> While training a novel safety-aligned model is outside the scope of this benchmark paper, the literature strongly supports our claim that training can bridge this gap. Works such as SafeWorld (Yin et al., 2024) and the Instruction Hierarchy (Wallace et al., 2024) demonstrate that targeted alignment and hierarchy prioritization can significantly improve adherence to safety constraints. Our results serve to highlight that current "off-the-shelf" agents lack this specific business-context alignment, identifying the gap that future training regimes must close.
>
>
> > Trade off between performance and confidentiality awareness
>
> We have expanded our analysis with claude-sonnet-4.5 and qwen3-coder, and the results reinforce our original conclusion. The trade-off remains the dominant trend across the most capable models. For instance, claude-sonnet-4.5, gpt-4o, gemini-2.5-pro, and qwen3-coder all exhibit decreased task completion when the confidentiality prompt is used. These findings confirm that for models effectively following safety instructions, simple prompting often incurs a quantifiable cost in task efficacy.
>
>
> > Intermediate metrics
>
> We appreciate the reference to $\tau^2$-Bench and the value of intermediate metrics for diagnostics. However, we deliberately chose not to enforce rigid intermediate assertions in CRMArena-Pro due to the high flexibility of our problem space. Unlike benchmarks with more linear workflows, tasks in our Salesforce environment often allow for multiple valid solution paths—for instance, an agent might retrieve the same information using a precise SOQL query, a broad SOSL search, or by traversing object relationships in different orders.
>
> Defining static natural language assertions for every possible valid trajectory would restrict the agents' autonomy and potentially penalize valid, but non-standard workflows. Given that the primary goal of an autonomous business agent is to achieve the correct business outcome regardless of the specific procedural steps taken, we prioritized outcome-based evaluation metrics (Exact Match and F1) to accommodate this realistic diversity in problem-solving strategies.
>
> > User simulator cost
>
> We appreciate the reviewer’s inquiry regarding the cost implications of using LM-simulators. We have estimated the total cost to run the full benchmark (4,280 queries) with the three user simulator models evaluated in Table 6 (gpt-4o-mini, gpt-4o, and o1).
>
> A rough breakdown cost for running the entire benchmark is as follows:
>
> - gpt-4o-mini: $4
>
> - gpt-4o: $60
>
> - o1: $360
>
> It is important to note that the cost of the user simulator is a small fraction of the agent's inference cost. Unlike the agent, which must process a massive prompt containing the full database schema (25 objects), action definitions, and tool outputs, the user simulator operates with a comparatively lightweight context (task goal and conversation history).

---

> ### Author Response · Authors · 2025-12-12
>
> > Elaborate on different tasks in the benchmark
>
> We appreciate the opportunity to clarify the distinction between the business scenarios. In CRMArena-Pro, the difference between Sales, Customer Service, and CPQ is primarily defined by the data context and domain knowledge required, while the operation types are defined by the business Skills.
>
> Specifically, the business scenarios dictate which subset of the schema the agent must navigate: Customer Service interacts with objects like Case and Solutions, Sales involves Lead and Opportunity, and CPQ involves intricate Quote and Pricebook structures. However, the type of reasoning required cuts across these domains. For example, both "Service Case Routing" and "Sales Lead Routing" fall under the Workflow Execution skill. They share the same operational requirement (applying conditional logic to route an item) but require accessing distinct data clusters and applying different business rules. Therefore, including all three domains ensures that the benchmark tests the agent's ability to generalize skills across the full, interconnected enterprise schema.

---

### Review · Reviewer_nEuw · 2025-10-23

**Summary Of Contributions:**

This paper presents CRMArena-Pro, a benchmark for evaluation of LLM agents focused on business use cases. This is an extension of an existing benchmark called CRMArena (Huang et al, 2025). CRMArena-Pro expands on CRMArena in the following ways: 1) introduces 19 new expert-validated tasks that cover B2B use cases including Sales and CPQ (configure, price, quote) processes, 2) incorporates multi-turn interactions in inputs, 3) conducts confidentiality aware assessments that evaluate the model’s ability to handle sensitive information appropriately.

Both CRMArena and CRMArena-Pro are grounded in the schema of the Salesforce Service Cloud, and evaluation data is generated synthetically and validated for quality. The quality validation involves deduplication to prevent redundant entries, format verification to ensure compliance with the schema, and content verification to ensure alignment within the simulated business context. CRMArena-Pro contains 4280 queries that cover 19 tasks, with additional queries for confidentiality aware assessment, which involves stress-testing the model to reveal private customer information, internal operational data or confidential company knowledge. Expert evaluation of the quality of the data with professionals recruited via the User Interviews platform showed that both the B2B data and B2C data is rated realistic or highly realistic >60% times.

The schema associated with the dataset can be accessed using a GUI and through an API. LLM agents are evaluated through the API, while expert studies are conducted using the GUI. Evaluation is done based on task complication efficacy (F1 score and exact match) and confidentiality awareness (measured by refusal rate using an LLM judge). Competitive LLM ReAct-based agents based on models such as o1 and gemini-2.5-pro achieve 58% on CRMArena-Pro in single-turn scenarios, and 35% in multi-turn scenarios, showing headroom for improvement. In terms of confidentiality evaluation, all models show near-zero refusal rate but using a system prompt with explicit instructions to decline requests for sensitive information, increases the refusal rate at the cost of lower task completion efficacy.

The paper does additional analyses to explore the tradeoff between performance and cost, and finds that Gemini-2.5-Pro provides a good tradeoff with low cost and high performance. They also provide actionable guidance for future work based on current failure modes of models.


Strengths:

- The paper is written well and really easy to follow. The main contributions of the paper and the differences from CRMArena are clear.
- The proposed benchmark appears to be fairly challenging for current LLM agents, which can encourage the adoption of this benchmark in the future. It contributes multi-turn data, and also does a confidentiality evaluation of agents, which are both grounded in real-world use cases of these agents.
- Experiments are conducted thoroughly, with sufficient experimentation of different models and exploration of various settings, all of which provide more backing to the claims made by the authors.
- The analysis is also pretty useful and touches upon interesting aspects of this evaluation, including cost-performance tradeoffs, multi-turn vs single-turn settings, failure modes of current models etc.


Weaknesses:

- While the expert validation of the quality of data is useful, there is little information about the subjectivity or inter-annotator agreement of the realisticness judgements from experts. This raises some concerns about the quality of the data, especially because ~62-67% of the data being rated realistic is not sufficiently high. I am also curious about what tasks are not rated as realistic, and if the authors considered just removing those tasks.
- There is little to no qualitative analysis grounded in real examples from the dataset provided in the paper. This would be really helpful to understand the format and content of the tasks.
- While the error analysis (“actionable guidance”) in section 4.4 is useful, this analysis would have been more useful if the authors conducted a systematic evaluation of the errors and presented the distribution of all the error types described here.
- There is no validation / human agreement reported for the LLM-as-a-judge evaluations. This is important to establish the reliability of the metrics that use LLM-as-a-judge.


Given the strengths of the paper and the amount of effort the above changes would take, I would recommend for acceptance.

**Audience:**

Yes

**Audience Explanation:**

The paper and the accompanying dataset could be useful for both industry professionals and academics, as it gets closer to evaluating LLM agents in realistic settings. There is a lot of headroom for improvement, which will encourage adoption.

**Claims And Evidence:**

Yes

**Claims Explanation:**

There is enough backing for the main claims made by the paper. However, please see my comments about LLM-as-a-judge and expert validation above.

**Requested Changes:**

- Validate the use of LLM-as-a-judge using human evaluation on a subset of data and report agreement rates.
- Please report the inter-annotator agreement rate for the realisticness labels from expert studies. If repetitions were not collected, please consider collecting repetitions on a small amount of data and reporting agreement numbers.
- More systematic error analysis, with a distribution of error types for the best performing models, would guide future work on which slices of the data to focus on.
- More qualitative examples, both in the dataset construction section and the discussion section would be useful.

I think all of these are doable within a short time window, so I would emphasize making these changes before the paper is accepted.

---

> ### Author Response · Authors · 2025-12-12
>
> We would like to thank reviewer nEuw for the positive assessment of our work, specifically for recognizing that the paper is well-written and easy to follow, and for appreciating the clarity of our contributions relative to the original CRMArena. We are also grateful for the reviewer's acknowledgement of CRMArena-Pro as a challenging and adoption-worthy benchmark that grounds multi-turn interactions and confidentiality evaluations in real-world use cases. We also appreciate the positive remarks regarding the thoroughness of our experiments and the utility of our analyses on cost-performance tradeoffs and failure modes. We have carefully considered the reviewer's constructive feedback regarding data validation, error analysis, and qualitative examples, and have addressed these points in detail below.
>
>
> > Expert study clarification
>
>
>
> Regarding inter-annotator agreement, we view realism as an inherently subjective measure heavily influenced by the specific industry background of each expert (e.g., a Sales Director vs. a Portal Administrator). Consequently, standard agreement metrics are less applicable than in objective labeling tasks. To provide full transparency, we present the complete distribution of ratings in Figure 4. As the figure illustrates, while there is natural variance based on professional experience, the consensus is strongly positive.
>
> Addressing the concern that 62-67% is "not sufficiently high", we respectfully note that this number represents strictly "Realistic" and "Very Realistic" ratings. Crucially, the majority of the remaining ratings were "Neutral" rather than "Unrealistic." In the context of our human study, the majority of these remaining ratings were "Neutral," defined in our study as "moderately realistic" (Appendix D). This reflects the subjective nature of the evaluation, where experts from different industries may have varying expectations for CRM data, rather than objective errors.
>
>
> Finally, regarding the suggestion to remove "unrealistic tasks," we wish to clarify a misunderstanding of the study design. As detailed in Appendix D and Figure 8, experts were asked to rate the realism of the Salesforce Objects (the underlying data schema and records), not the specific tasks or queries. The tasks themselves (e.g., Lead Qualification, Case Routing) are directly sourced from standard business workflows and day-to-day operations of CRM professionals, ensuring their inherent realism. Therefore, there were no specific "unrealistic tasks" identified for removal.
>
>
> > Error type distribution
>
> We agree that concrete examples and systematic qualitative analysis are essential for understanding the benchmark's complexity and the nature of agent failures. In the revised manuscript, we have expanded Section 4.4 to include a detailed error analysis grounded in 50 randomly sampled failure cases from the top-performing agent, gemini-2.5-pro.
>
> Our analysis reveals a distinct distribution of failure modes in single-turn settings:
>
>
> - Schema & Domain Knowledge Gaps (32%): Agents frequently confuse semantically similar objects, such as querying PricebookEntry instead of OrderItem, or misinterpreting Lead versus Opportunity stages.
>
>
> - Tool & Syntax Proficiency (28%): Despite high general capability, agents often struggled with enterprise-specific grammar, producing invalid SOQL queries or selecting SOQL when a text-based SOSL search was more appropriate.
>
>
>
> - Reasoning Deficits (24%): In these cases, the agent successfully retrieved the correct Knowledge Article or record but failed to extract the specific clause or insight required to answer the user’s question.
>
>
> - Ambiguity Detection Failures (16%): Agents often hallucinated missing details or assumed default values rather than flagging underspecified inputs.
>
> For multi-turn interactions, we further sampled additional 50 erroneous trajectories and quantified that 45% of failures stemmed from premature finalization, where the agent attempted to answer immediately rather than utilizing the clarification-seeking action to acquire necessary, withheld information.
>
> > More qualitative examples
>
> Thank you for the suggestion. We have added more qualitative examples in Section 3.2 and Section 4.4.

---

> ### Author Response · Authors · 2025-12-12
>
> > LLM-as-a-Judge reliability
>
> As discussed in Section 4.4, our evaluation is explicitly designed to be a constrained, binary classification rather than a subjective open-ended assessment. The judge is prompted with specific refusal phrases and instructed to output a simple "YES" or "NO" based solely on whether the agent explicitly refused the sensitive query. This reduces the potential for biases such as verbosity or self-enhancement often associated with LLM judges, making it a robust metric for this specific type of safety check.
>
> To empirically validate this reliability, we conducted a manual inspection of 50 randomly sampled judge outputs against the agent trajectories. Our manual evaluation identified only a single error, confirming that the judge aligns closely with human ground truth for this specific refusal-detection task.

---

### Review · Reviewer_rVEj · 2025-12-09

**Summary Of Contributions:**

This paper introduces a large-scale CRM benchmark covering service, sales, and Configure–Price–Quote (CPQ), with 19 tasks mapped into four business skill categories. It includes sizable synthetic B2B and B2C datasets and represents a broader and more realistic setup than prior CRM benchmarks. Beyond task coverage, the benchmark explicitly evaluates multi-turn interactions and confidentiality handling, which are critical enterprise concerns and largely missing from existing benchmarks. The results show that current models still struggle on these aspects, making this a challenging and useful resource for the agentic AI community.

## Strength

- I want to highlight the overall scope. The benchmark is comprehensive: it covers service, sales, and Configure, Price, Quote (CPQ), and the 19 tasks map cleanly into the four business skill categories. The dataset size is sufficiently large: 29,101 and 54,569 records for the B2B and B2C orgs, respectively. It’s a much broader setup than previous CRM benchmarks. The merged schema and the size of the synthetic data also show a fair amount of engineering work overall. I believe the community will find this resource useful, and in my understanding, there are not many realistic CRM-style environments that exist.
- Another strength is the confidentiality aspect (handling sensitive information and appropriate data protocols). The authors break this into three categories (private customer info, internal operational data, confidential company knowledge) and evaluate it directly. This is something enterprise setups actually probably care about (due to private knowledge). Further, as argued by the authors, the multi-turn design is missing from existing benchmarks such as WorkArena++. Typical workflow is supposedly multi-turn, so this is good for realistic benchmarking, certainly (It’s closer to how actual workflows happen).
- I also like the rigorous analysis and reporting: variance across runs, cost/latency numbers, and the clarifications with sub-questions. I especially enjoyed the discussion on the trade-off between confidentiality vs performance in Section 4.3.
- Overall, the benchmark tasks are quite challenging for state-of-the-art models in some aspects, such as multi-turn and confidentiality, which matter most for a new benchmark.

## Weakness
- I have a few questions and concerns about the design choices. First, while CRMArenaPro is tailored to B2B/B2C workflows, there’s no real justification for picking Salesforce specifically. Is it just the most convenient sandbox, or is there a stronger claim about representativeness? Second, the setup is entirely API-based, even though a lot of real CRM usage is GUI-driven. Some discussion of what is gained and what is lost by focusing on SOQL/SOSL over GUI workflows would help. Right now, several of these design decisions feel under-motivated.
- Another concern I have is the quality of the benchmark data. I appreciate that there is an expert study, but the realism numbers look modest: only about two-thirds of ratings are “realistic or very realistic,” and Figure 4 shows a lot of mass in neutral or even unrealistic buckets. In the original CRMArena paper, the authors reported something closer to ~90% “realistic or better,” which feels more like the right bar. It’s not clear why lower-quality parts of the data weren’t filtered out, or at least split off, so that users can work with a “realistic-only” subset. For a benchmark that is supposed to guide future work, data quality should be treated as a first-class concern.
- The prompting/scaffold setup also feels a bit narrow. The current system prompt assumes the model knows how to use SOSL and SOQL and gives some tool-level guidance, but it sits in the middle: not minimal, not full access. Since scaffolds vary widely in practice, it would be useful to see ablations at both extremes: (a) almost no tool guidance, (b) full schema + API documentation, and (c) a small library of SOQL/SOSL patterns. That would make it easier to tell how much of the difficulty comes from missing enterprise knowledge vs actual reasoning/task execution failures. This relates to the author’s discussion on page 11, “What are the actionable guideline for closing the gaps?”  The author is suggesting that fine-tuning is needed to learn the enterprise-specific grammar of SOQL and SOSL. I think with current models having huge context, a lot of this should be injected into the model just in inference time. I’m curious what authors think of that or whether they considered it.
- I’d also like more concrete task examples in the main paper (probably in the style of infographics, which is easier to follow). Even though the domain is specific (which makes it more important to give clear examples for the readers), a few fully worked instances per skill (user query, tool calls, final answer, and what is considered correct) would be easier to follow. Currently, it’s a lot of dense information to keep track of, which could be better communicated either with examples or infographics. Right now, the description is fairly abstract, and you have to reverse-engineer a lot of details from the prose and tables.
- Regarding result analysis, it mostly feels like just spelling out the numbers rather than proper analysis.
    - For example, in the paragraph titled *“Performance varies notably across business skills, with Workflow Execution often proving more tractable in single-turn settings.”* The author could have commented on why the task “Workflow Execution” is easier than others. What are the specific properties of that task which lead to this behaviour? I am not listing all of them, but the discussion should be more nuanced to better interpret the reported results (as applicable).
   - Another example, *“LLM agents face substantial challenge in acquiring additional information through clarification.”* The author describes a phenomenon, but doesn’t give a clear example (or data point) for the reader to actually understand and verify it. Please give an example of the analysis done (referring to appendix is fine).


- Finally, the evaluation for confidentiality (and some tasks) relies on an LLM-as-judge setup. The authors argue it’s a simple “refusal vs not” classification, which helps, but I would still expect at least a small human-validated subset reporting judge–human agreement, just to show that this piece of the pipeline is reliable.

**Audience:**

Yes

**Audience Explanation:**

I believe the agentic-LLM (AI) community at large would be interested. Given the widespread adoption of LLMs across many business applications, this work seems quite useful. One minor concern is the choice to focus exclusively on Salesforce; I would like the authors to justify this decision and discuss how generalizable the benchmarks and the findings are beyond this specific enterprise setting.

**Broader Impact Concerns:**

The paper contains a broader impact section, and I don't have any concerns.

**Claims And Evidence:**

Yes

**Claims Explanation:**

Yes, the authors propose a benchmark and provide a detailed evaluation on different aspects of the benchmark. The benchmark substantially contains new tasks as claimed by the authors and also shown to be challenging for state-of-the-art models.

**Requested Changes:**

I have a few questions where I’d like additional clarification, as well as follow-ups on some of the weaknesses I mentioned earlier. For my recommendation of acceptance, data quality and the presentation of clearer, more readable examples are the most critical aspects.

## Questions
- My understanding is that the agent action space is essentially “read-only querying + respond,” with no record creation/updates. Does that under-represent actual CRM work, where editing records and driving workflows are central? What exact slice of CRM work is this benchmark meant to cover?
- How many tool calls are actually needed to solve a task in practice? Are most instances solvable with a single SOQL/SOSL query, or do successful trajectories usually require multiple calls? In the multi-turn setting, what are the average number of turns and average number of tool calls per task (and how does that differ by skill)?
- What is the random-chance or trivial baseline for each task type? For example, routing and approval tasks have a fixed number of candidate queues/decisions; numeric tasks could be compared against very simple heuristics.
- On Salesforce and generalizability: can the authors comment on what they expect to transfer cleanly to other CRMs (e.g., task types, skill definitions, multi-turn patterns) and what is Salesforce-specific (schema details, SOQL/SOSL, CPQ object conventions)? Some explicit discussion here would help users judge how “CRM-wide” this benchmark really is.

---

> ### Author Response · Authors · 2025-12-12
>
> We express our gratitude to reviewer rVEj for their positive assessment of the comprehensive scope and engineering effort behind CRMArena-Pro. We are encouraged by the reviewer's recognition that our benchmark's breadth—covering Service, Sales, and CPQ across nearly 85,000 combined records—fills a critical gap in realistic CRM environments. We also appreciate the reviewer highlighting the value of our confidentiality framework and multi-turn design as essential components for enterprise-grade evaluation. Finally, we thank the reviewer for acknowledging the rigor of our analysis, particularly regarding stability, cost-latency trade-offs, and the nuanced performance-confidentiality dynamics. We have provided detailed responses to the reviewer's specific questions and concerns below.
>
> > Choice of Salesforce
>
> Our decision to use the Salesforce CRM system was driven by representativeness rather than mere convenience. Salesforce holds the largest market share in the CRM space and effectively defines the standard data model for B2B operations. By merging schemas from Service, Sales, and CPQ Clouds to create a complex environment of 25 interconnected objects, we model the high-fidelity data relationships (e.g., Opportunity to Quote to PricebookEntry) that define modern enterprise complexity. While the schema is Salesforce-specific, the underlying workflow understanding and execution challenges, such as determining lead qualification criteria or navigating approval policies, are universal to enterprise software.
>
>
> > API over GUI workflows
>
> Regarding the focus on API over GUI workflows, we argue that APIs are a more practical interface for the complex, data-intensive tasks defined in CRMArena-Pro. Many tasks, such as Handle Time Understanding, require filtering records by specific time periods, computing independent averages for multiple users, and ranking the results to determine minima or maxima. Performing these multi-step computations via a standard GUI is extremely long horizons and prohibitively challenging even for human operators, who would typically rely on backend reporting tools or exports to spreadsheets. Therefore, focusing on APIs is more reasonable, as it enables the agent to perform the requisite aggregations efficiently and ensures the benchmark evaluates business reasoning and calculation capabilities rather than the mechanical bottlenecks of visual navigation.
>
>
> > Data realism concerns
>
> We appreciate the close examination of our data quality metrics. Addressing the concern that 62-67% is "modest" compared to the original CRMArena, we respectfully note that this figure represents strictly "Realistic" and "Very Realistic" ratings. Crucially, the majority of the remaining ratings were "Neutral" rather than "Unrealistic." In the context of our study (Appendix D), "Neutral" is defined as "moderately realistic," representing functional data that may simply lack specific nuances familiar to an expert's particular niche. The drop from ~90% in the original CRMArena to ~67% here reflects the __significantly increased complexity of CRMArena-Pro__, which merges Sales, Service, and CPQ schemas into a 25-object web. As the schema complexity and inter-connectivity increase, experts naturally apply more conservative ratings based on their highly specialized B2B experiences.
>
>
> Regarding the suggestion to "filter out" lower-rated parts, this is structurally infeasible due to the highly interconnected nature of the Salesforce schema. As shown in the dependency graph in Figure 37, objects like Account or Opportunity serve as foundational nodes with foreign keys linking to almost every other object (e.g., Cases, Quotes, Contacts). Experts provided ratings at the object level, not the record level. Therefore, removing an object that received a "Neutral" rating would break the referential integrity of the entire database, rendering the environment non-functional.

---

> ### Author Response · Authors · 2025-12-12
>
> > Prompt design and long-context inference
>
> We appreciate the reviewer’s suggestion to explore the boundaries of prompt scaffolding. First, we wish to clarify that our current system prompt does include the full database schema (all 25 objects and their relationships), as detailed in Appendix E.1 and Figure 9. This design was chosen to align with the standard ReAct framework, ensuring our results are comparable to established agentic baselines rather than being specific to a novel, highly-optimized scaffold.
>
> Regarding the hypothesis that long-context inference effectively replaces fine-tuning: While extensive context helps with basic syntax, our error analysis in Section 4.4 reveals that agent failures often stem from deeper __"gaps in schema and domain knowledge"__ and __"weaknesses in reasoning over retrieved content"__. For instance, agents frequently confuse conceptually similar objects (e.g., Lead vs. Opportunity) or fail to detect ambiguity, even when the definitions are present in the context window. Therefore, our recommendation for fine-tuning is not merely about syntax, but about helping agents "internalize standard operating workflows"  and business logic—capabilities that often require training to stabilize beyond what inference-time documentation can achieve.
>
>
> > Concrete examples
>
> Thank you for the suggestion. We have expanded our section  3.2 and 4.4 to include more examples.
>
>
> > Result analysis
>
> We agree that adding causal interpretation would strengthen the paper. In the revised manuscript, we have expanded the discussion to explicitly address the "why" behind these performance trends in Section 4.2.
>
> First, we clarify that its higher tractability stems from its explicit, rule-based nature. As defined in Section 3.2, tasks like Service Case Routing provide the agent with a clear, textual policy (e.g., "Assign to Agent X if Condition Y is met") directly in the context. Unlike Database tasks, which are prone to "limited tool and syntax proficiency" errors in SOQL generation, or Text tasks requiring synthesis of unstructured corpora, Workflow tasks primarily require logical rule application, a capability where current reasoning models excel.
>
> Additionally, we have added a concrete breakdown of the failure modes to Section 4.2. Specifically, our manual analysis of 20 failed trajectories revealed that 45% (9/20) of failures resulted from the agent failing to acquire necessary, withheld information. For example, in a Sales Lead Routing task where the routing policy depends on the lead's region, the simulated user provides a lead without a region. Instead of asking "What region is this lead in?", the agent "prematurely finalizes" the task by hallucinating a region or assigning a default agent, illustrating the specific inability to recognize ambiguity.
>
> > LLM-as-a-Judge Reliability
>
> We appreciate the reviewer’s emphasis on validating automated metrics. As discussed in Section 4.4, we designed the confidentiality evaluation as a highly constrained binary classification task rather than a subjective quality assessment. The judge is explicitly prompted to output "YES" or "NO" based on the presence of clear refusal signals (e.g., "I cannot provide that information"), minimizing the ambiguity often associated with LLM judges.
>
> To empirically validate this reliability, we conducted a manual inspection of 50 randomly sampled judge outputs against the agent trajectories. Our review identified only a single discrepancy (an error rate of 2%), confirming that the judge aligns closely with human ground truth for this specific refusal-detection task.
>
>
> > Read-only action space concern
>
> We appreciate the reviewer’s thoughtful question regarding the scope of CRM work. We clarify that CRMArena-Pro is specifically designed to evaluate the Analysis, Retrieval, and Decision-Making "slice" of CRM operations, which we identify as the primary cognitive bottleneck for current LLM agents.
>
> While real-world CRM work involves executing updates (e.g., clicking "Save" or running an UPDATE query), the core difficulty lies in the reasoning that precedes that action. For instance, determining which sales rep qualifies for a lead (Lead Routing) or identifying why a quote violates policy (Policy Compliance). Once the correct decision is made, the mechanical act of creating or updating a record is a trivial operational step. Therefore, we treat the agent's "Respond" output as the verified intent to execute that workflow. This design choice allows us to rigorously evaluate the agent's reasoning capabilities without the significant overhead and instability associated with managing stateful environment resets and side effects in a benchmark setting.

---

> ### Author Response · Authors · 2025-12-12
>
> > Tool call details
>
> The number of tool calls vary by task. In general, each task takes about 3-7 correct tool calls. As for the average number of turns/clarifications, we detail this in Figure 6.
>
> > Random baseline performance
>
> We clarify that random-chance performance varies significantly by task type due to the diversity of the output spaces. For retrieval and routing tasks (e.g., Service Case Routing, Lead Routing), the search space is extensive, selecting the correct Agent ID from over 200 users or the correct Product ID from dozens of options, resulting in a random-chance probability of less than 0.5%. Similarly, temporal analysis tasks (e.g., Monthly Trend Analysis) have a 1/12 baseline. Given these large search spaces and the requirement for exact ID matching or correct reasoning, trivial heuristics yield near-zero success rates across the benchmark.
>
>
> > Transferability of the benchmark
>
> We would like to distinguish between the cognitive/operational requirements (which are universal) and the implementation details (which are specific).
>
> - Transferable Elements: The core Business Skills defined in Section 3.2, Workflow Execution, Policy Compliance, and Textual Reasoning, etc are fundamental to all major CRM platforms (e.g., Microsoft Dynamics 365, HubSpot, Zoho). For instance, the logic required to "route a case based on agent availability" or "validate a quote against a discount policy" remains identical regardless of the underlying database. Similarly, the multi-turn dynamics (clarifying ambiguity) and confidentiality protocols (protecting PII) are platform-agnostic challenges essential for any enterprise agent.
>
> - Salesforce-Specific Elements: The specific the query syntax (SOQL/SOSL) is naturally specific to Salesforce.
>
> However, we argue that the benchmark effectively acts as a proxy for "enterprise understanding/execution capability." An agent that can successfully navigate the complex, interconnected schema of CRMArena-Pro demonstrates the capacity to learn and reason over any arbitrary enterprise data model, making the findings highly relevant beyond the Salesforce ecosystem.

---

### Author Response · Authors · 2026-01-18

Dear Action Editor and Reviewers,

We are writing to politely follow up on our revised paper and author response submitted over a month ago. We appreciate the detailed feedback you provided and have worked to address all concerns comprehensively.

Please let us know if any further clarifications are needed from our side.

Thank you for your time and continued consideration.

Best regards,

The Authors

---

> ### Comment · Action_Editor_aKqF · 2026-01-18
> **Waiting for final recommendation**
>
> Hi,
> We are still waiting for one final reviewer recommendation which I think we will be able to get shortly.
> Thanks.

---

> > ### Author Response · Authors · 2026-01-19
> >
> > Thank you so much for the update and for managing our submission!

---

### Decision · Action_Editor_aKqF · 2026-01-26

**Recommendation:** Accept as is

**Audience:**

Yes

**Audience Explanation:**

Both agents and multi-turn performance are of wide interest to the community at the moment, so this seems like a benchmark that will interest many.

**Claims And Evidence:**

Yes

**Claims Explanation:**

The authors present an interesting benchmark that tests the ability of agents to handle real life business scenario in a multi-turn setup. They thoroughly evaluate models, adding models based on reviewer request, and do a thorough analysis of the bechmark and the empirical findings.

Both agents and multi-turn performance are of wide interest to the community at the moment, so this seems like a benchmark that will interest many.